# Specializing Versatile Skill Libraries using Local Mixture of Experts

**Onur Celik**[1], **Dongzhuoran Zhou**[1], **Ge Li**[1], **Philipp Becker**[1], and **Gerhard Neumann**[1]

[1]Autonomous Learning Robots, KIT, Germany
{celik, ge.li, philipp.becker, gerhard.neumann}@kit.edu
dongzhuoran.zhou@outlook.com

**Abstract:** A long-cherished vision in robotics is to equip robots with skills that match the versatility and precision of humans. For example, when playing table tennis, a robot should be capable of returning the ball in various ways while precisely placing it at the desired location. A common approach to model such versatile behavior is to use a Mixture of Experts (MoE) model, where each expert is a contextual motion primitive. However, learning such MoEs is challenging as most objectives force the model to cover the entire context space, which prevents specialization of the primitives resulting in rather low-quality components. Starting from maximum entropy reinforcement learning (RL), we decompose the objective into optimizing an individual lower bound per mixture component. Further, we introduce a curriculum by allowing the components to focus on a local context region, enabling the model to learn highly accurate skill representations. To this end, we use local context distributions that are adapted jointly with the expert primitives. Our lower bound advocates an iterative addition of new components, where new components will concentrate on local context regions not covered by the current MoE. This local and incremental learning results in a modular MoE model of high accuracy and versatility, where both properties can be scaled by adding more components on the fly. We demonstrate this by an extensive ablation and on two challenging simulated robot skill learning tasks. We compare our achieved performance to LaDiPS and HiREPS, a known hierarchical policy search method for learning diverse skills.

**Keywords:** Curriculum, Versatile Skill Learning, Episodic Hierarchical RL

## 1 Introduction

Human motor skills are precise and versatile, which allows us to perform motor tasks in different ways while achieving a consistent movement quality. For example, when playing table tennis, we can hit the ball in various ways while still targeting a specific landing point of the ball on the opponent's side of the table. Another example is grasping an object which lies behind an obstacle. We can grasp even small objects precisely with different grasp types while avoiding the obstacle. Such versatile skills are crucial if we want to employ robots in unstructured and dynamically changing environments. Such skills, often represented as movement primitives, were already successfully learned for challenging robot learning tasks, e.g., the ball-in-a-cup [1, 2] task, by a variety of policy search algorithms [3]. Yet, most of these algorithms cannot find multiple, versatile, and precise solutions to the multi-modal solution space, as they usually assume a Gaussian policy [4, 2, 3, 5].

In this paper, we model versatile behavior with contextual skill libraries of motion primitives [6, 7], formalized by Mixtures of Experts (MoEs). Here the context defines task properties, e.g., reaching different goal positions or different friction parameters of an object to manipulate [5]. Our goal is to learn versatile skills, i.e., different movement styles to solve a given context. Given a context, the MoE first selects a component, i.e., a motion primitive, to execute. Subsequently, the component adjusts the primitive's parameters and a controller executes the primitive. Such models are

5th Conference on Robot Learning (CoRL 2021), London, UK.

already commonly used [8, 9]. Yet, the quality and versatility of the learned skill libraries remain limited using existing algorithms. Most algorithms are based on expectation-maximization (EM) [10] techniques [8], and suffer from local optima and mode averaging [10] which prevents the single components from specializing in a local context region. Moreover, existing algorithms [8, 9] do not explicitly optimize the versatility of the library. Hence, they often yield degenerated libraries with only a single movement style. We propose a new objective for learning contextual, precise, and versatile MoE models based on a maximum entropy formulation. We also introduce a learnable context distribution, which provides a curriculum for each component of the MoE model. We use a variational lower bound [11] to decompose the objective into individual updates for the components and their related local context distributions, allowing the components to specialize in local regions of the context space and preventing mode averaging. Due to the curriculum, the MoE does not have to cover the whole context space during training, which prevents the averaging effects that negatively affect most other approaches. Yet, not covering the whole context space leads to poor performance for some contexts, which is also undesirable. Thus, we introduce a heuristic-free mechanism to add new components during training until the whole context space is covered. Hence our algorithm provides a modular approach that learns highly precise and versatile skills that cover the whole context space.

We evaluate our approach in a simulated beer pong and a table tennis environment. Both environments allow different motion styles to solve the tasks, which are discovered by our algorithm. Moreover, we present ablation studies showing the importance of the single elements and hyperparameters of our algorithm.

## 2   Related Work

**Contextual Episodic Policy Search.** Episodic policy search [3] aims at maximizing the expected return by optimizing the parameters $\boldsymbol{\theta}$ of a controller, e.g., a motion primitive [12, 6]. Most approaches use a stochastic search distribution $\pi(\boldsymbol{\theta})$ over the parameter space and aim to optimize the expected return under this search distribution [3], i.e.,

$$\max_{\pi(\boldsymbol{\theta})} \mathbb{E}_{\pi(\boldsymbol{\theta})}[R(\boldsymbol{\theta})],$$

where $R(\boldsymbol{\theta}) = \sum_t r_t$ is the summed reward over a whole episode obtained when executing controller parameter $\boldsymbol{\theta}$. As it is common in the literature, we will denote $\pi(\boldsymbol{\theta})$ as our policy even though it only indirectly chooses the control actions of the agent by selecting the controller parameters $\boldsymbol{\theta}$. Different optimization schemes such as policy gradients Sehnke et al. [13], natural gradients [14], stochastic search strategies [15, 16] or trust-region optimization techniques [8, 17] have been used. Researchers extended these approaches to the contextual setting [18, 4], where the search distribution $\pi(\boldsymbol{\theta}|\mathbf{c})$ now depends on a context vector $\mathbf{c}$ which describes the task, e.g., a goal location to reach. The contextual objective is typically formalized as

$$\max_{\pi(\boldsymbol{\theta}|\mathbf{c})} \mathbb{E}_{p(\mathbf{c})} \left[ \mathbb{E}_{\pi(\boldsymbol{\theta}|\mathbf{c})}[R(\boldsymbol{\theta}, \mathbf{c})] \right],$$

where $p(\mathbf{c})$ is the given distribution over context vector and the rewards now also depends on the context. Klink et al. [2] introduce a curriculum into contextual policy search. By having an adaptable distribution over the contexts, they allow the agent to concentrate on easy-to-solve contexts first and then generalize to the whole context space.

**Versatile Skill Learning.** The Hierarchical Relative Entropy Policy Search (HiREPS) algorithm [8] extends the classical Relative Entropy Policy Search (REPS) approach [19] to MoEs, which allows learning versatile skills in a contextual episodic policy search setting. In a similar approach, Layered Direct Policy Search (LaDiPS) [9] also uses MoE policies, but builds on Model-Based Relative Entropy Policy Search (MORE) [17] instead of REPS. Both HiREPS and LaDiPS address the same problems as our approach, yet there are also considerable differences. First, HiREPS jointly optimizes the whole mixture model and introduces an additional constraint, which bounds the entropy loss of the responsibilities in each iteration. This constraint is crucial for obtaining versatile and well-performing solutions. LaDiPS uses separate updates for the different parts of the mixture but also relies on additional constraints, where the entropy of the gating is lower bounded with a constant value. In contrast, for our approach, the objective and separate updates of the individual mixture parts follow naturally from the maximum entropy formulation. Second, neither HiREPS

nor LaDiPS uses a curriculum for training. Thus, in both approaches, the MoE always has to cover the whole context space and, hence, the components cannot specialize.

**Variational Inference.** Our algorithm is also related to several recent advances in variational inference [11, 20, 21]. It is well known that maximum entropy RL is equivalent to inference in an appropriate probabilistic model [22]. Similar to previous works [23, 24], we exploit this relation and draw inspiration from recent research into variational inference and density estimation for Gaussian mixture models and MoEs [11, 20, 21]. We reformulate the lower bound objectives introduced in those approaches for our maximum entropy RL setting and extend them with a curriculum for the mixture components.

**Related Step-Based Approaches.** In the step-based setting, the policy does not learn a function from contexts to parameters of an episodic controller but directly maps from system states to actions and the policy updates are performed with the information from each time-step. Practitioners often use deep neural networks to parameterize step-based policies, giving rise to the field of deep RL. In this context, versatile policy learning is also a very active research area [25, 26, 27, 28]. These approaches use a similar MoE model where the mixture component is only chosen at the beginning of an episode. Yet, the component is chosen randomly without conditioning on a context or state variable. Moreover, these approaches reformulate a mutual information based objective into a maximum entropy objective while we develop a more direct maximum entropy maximization.

**Curriculum Learning.** Researchers also worked on introducing curricula into deep RL. In a first approach Ghosh et al. [29] proposed partitioning the initial state distribution using clustering. They then learn individual policies for each partition while keeping the partitioning fixed. Strictly, this is not a curriculum as the partitioning is not adjusted, yet it still allows specialization of the individual policies in different regions of the state space. To automatically generate and adapt a curriculum for deep RL approaches, Klink et al. [30] extended their approach from the episodic setting [2] to the step-based setting. Yet, neither of these approaches addresses versatility. While we follow an episode-based approach, both methodologies have their benefits and limitations [3] which are, however, not the focus of this paper. We offer further discussion and quantitative comparison to a step-based approach for a common benchmark task in the appendix, therefore.

**Options.** A related hierarchical approach is the options framework [31, 32, 33]. The options framework extends the standard MDP to a semi-MDP to include a temporal abstraction of low-level control policies. Given a termination condition, the executed low-level policy can be terminated and another can be turned on. Our policy structure can be seen as a simplification of the options framework where the option is only selected at the beginning of each episode. Yet, the options framework does not explicitly address learning versatile skills.

## 3  Specializing Versatile Mixture of Expert Models

To allow versatile solutions, we employ a Mixture of Experts (MoE) model as policy representation which is given as $\pi(\boldsymbol{\theta}|\mathbf{c}) = \sum_o \pi(o|\mathbf{c})\pi(\boldsymbol{\theta}|\mathbf{c}, o)$, where $\pi(o|\mathbf{c})$ is the gating distribution, assigning a probability to component $o$ given the context $\mathbf{c}$ and $\pi(\boldsymbol{\theta}|\mathbf{c}, o)$ is the component distribution for component $o$, which adapts the motion primitive's parameters $\boldsymbol{\theta}$ to the given context $\mathbf{c}$. In this section, we derive a lower bound to optimize each component and its corresponding context distribution independently.In order to implement a curriculum over the context $\mathbf{c}$, we also introduce a learned context distribution $\pi(\mathbf{c}) = \sum_o \pi(\mathbf{c}|o)\pi(o)$, which is also a mixture model specified by the component-wise context distribution $\pi(\mathbf{c}|o)$ and the component weights $\pi(o)$. By applying Bayes' Rule to replace the gating distribution $\pi(o|\mathbf{c})$, we can now rewrite the general mixture of experts model as

$$\pi(\boldsymbol{\theta}|\mathbf{c}) = \sum_o \frac{\pi(\mathbf{c}|o)\pi(o)}{\pi(\mathbf{c})}\pi(\boldsymbol{\theta}|\mathbf{c}, o). \tag{1}$$

This policy definition allows each component $o$ to adjust its curriculum by explicitly optimizing for $\pi(\mathbf{c}|o)$ (Section 3.3) and thus concentrating on a local region in the context space. We model the components $\pi(\boldsymbol{\theta}|\mathbf{c}, o)$ as a linear conditional Gaussian distribution and the component-wise state distribution $\pi(\mathbf{c}|o)$ as a Gaussian. The prior weights $\pi(o)$ define a categorical distribution. Throughout the next sections, we denote every probability distribution which is adaptable through the optimization process with $\pi$ as it is part of our policy $\pi(\boldsymbol{\theta}|\mathbf{c})$. Furthermore we show the full derivations for the next sections in the Appendix A.

## 3.1 Maximum Entropy Skill Learning with Curriculum

We consider a maximum entropy objective [22] for episodic policy search, i.e.,

$$\max_{\pi(\boldsymbol{\theta}|\mathbf{c})} \mathbb{E}_{p(\mathbf{c})} \left[ \mathbb{E}_{\pi(\boldsymbol{\theta}|\mathbf{c})} \left[ R(\mathbf{c}, \boldsymbol{\theta}) \right] + \alpha H \left[ \pi(\boldsymbol{\theta}|\mathbf{c}) \right] \right], \tag{2}$$

where $p(\mathbf{c})$ is the task specific context distribution, $R(\mathbf{c}, \boldsymbol{\theta})$ is the reward function, $H(\pi(\boldsymbol{\theta}|\mathbf{c})) = -\int_{\boldsymbol{\theta}} \pi(\boldsymbol{\theta}|\mathbf{c}) \log \pi(\boldsymbol{\theta}|\mathbf{c}) d\boldsymbol{\theta}$ the entropy and $\pi(\boldsymbol{\theta}|\mathbf{c})$ is our MoE model. The reward maximization enforces preciseness while the entropy bonus enforces versatility. However, the standard maximum entropy objective does not allow each mixture component to create its own curriculum, since an optimization over the component-wise context distributions $\pi(\mathbf{c}|o)$ is not given. Inspired by the work from Klink et al. [2], we extend and modify the objective to

$$\max_{\pi(\boldsymbol{\theta}|\mathbf{c}),\pi(\mathbf{c})} \mathbb{E}_{\pi(\mathbf{c})} \left[ \mathbb{E}_{\pi(\boldsymbol{\theta}|\mathbf{c})} \left[ R(\mathbf{c}, \boldsymbol{\theta}) \right] + \alpha H \left[ \pi(\boldsymbol{\theta}|\mathbf{c}) \right] \right] - \beta KL \left( \pi(\mathbf{c}) \parallel p(\mathbf{c}) \right), \tag{3}$$

where $\alpha$ and $\beta$ are scaling parameters, $H(\pi(\boldsymbol{\theta}|\mathbf{c})) = -\int_{\boldsymbol{\theta}} \pi(\boldsymbol{\theta}|\mathbf{c}) \log \pi(\boldsymbol{\theta}|\mathbf{c}) d\boldsymbol{\theta}$ is the entropy and $KL \left( \pi(\mathbf{c}) \parallel p(\mathbf{c}) \right) = \int_{\mathbf{c}} \pi(\mathbf{c}) \log \frac{\pi(\mathbf{c})}{p(\mathbf{c})} d\mathbf{c}$ denotes the KL-divergence. Note the difference in the optimization variables compared to Eq. (2). The Kullback-Leibler (KL) term ensures that the context distribution $\pi(\mathbf{c})$ is close to the task specific context distribution $p(\mathbf{c})$ while $\pi(\mathbf{c})$ can choose to have low probability in regions of the context space where the MoE model is performing poorly. Note that this objective is similar to the negative I-projection of the joint distribution $\pi(\mathbf{c}, \boldsymbol{\theta})$ used in variational inference. Here, we also exploit the properties of the I-projection for learning the context distribution – the I-projection is mode seeking instead of mode-averaging and therefore allows for specialization on a local context area. Yet, the given objective is difficult to optimize for mixture models as the sum over the mixture components is appearing inside the log terms of the entropy and the KL. However, similar to [11], we can replace $\pi(\boldsymbol{\theta}|\mathbf{c})$ in Obj. (3) with our mixture model and apply Bayes theorem to arrive at

$$\max_{\pi(\mathbf{c},\boldsymbol{\theta})} \mathbb{E}_{\pi(o),\pi(\mathbf{c}|o)} \left[ \mathbb{E}_{\pi(\boldsymbol{\theta}|\mathbf{c},o)} \overbrace{\left[ R(\mathbf{c},\boldsymbol{\theta}) + \alpha \log \pi(o|\mathbf{c},\boldsymbol{\theta}) \right]}^{\text{augmented reward for component } o} + \overbrace{\beta \log p(\mathbf{c}) + (\beta - \alpha) \log \pi(o|\mathbf{c})}^{\text{augmented reward for context distributions}} \right] \tag{4}$$

$$+ \alpha \mathbb{E}_{\pi(o),\pi(\mathbf{c}|o)} \left[ H \left[ \pi(\boldsymbol{\theta}|\mathbf{c},o) \right] \right] + \beta \mathbb{E}_{\pi(o)} \left[ H \left[ \pi(\mathbf{c}|o) \right] \right] + \beta H \left[ \pi(o) \right].$$

The exact derivations are given in the Appendix A. Note that Eq. (4) is equivalent to Eq. (3), yet, instead of the entropy for the whole mixture model, it now contains entropy terms for each hierarchy layer of the MoE model, i.e., $H \left[ \pi(\boldsymbol{\theta}|\mathbf{c},o) \right]$, $H \left[ \pi(\mathbf{c}|o) \right]$ and $H \left[ \pi(o) \right]$, which are much simpler to compute. We also introduced log responsibilities $\pi(o|\mathbf{c},\boldsymbol{\theta}) = \pi(\boldsymbol{\theta}|\mathbf{c},o)\pi(o|\mathbf{c})/\pi(\boldsymbol{\theta}|\mathbf{c})$ and $\pi(o|\mathbf{c}) = \pi(\mathbf{c}|o)\pi(o)/\pi(\mathbf{c})$, occurring in the augmented rewards. They return a high negative reward for component $o$, if the context-parameter pair $(\mathbf{c},\boldsymbol{\theta})$ or the context sample $\mathbf{c}$ is already covered by another component, pushing the component to uncovered regions of the parameter space or context space respectively. Yet, the regions for the components will still overlap due the entropy bonuses for $\pi(\boldsymbol{\theta}|\mathbf{c},o)$ and $\pi(\mathbf{c}|o)$. Without this reward augmentation, each component could be optimized completely independently in Eq. (4) using a max-entropy objective. However, in this case, all components would concentrate on learning the best solution irrespective of whether this solution has already been covered by another component. Yet, the log responsibilities still hinder us from optimizing each component $\pi(\boldsymbol{\theta}|\mathbf{c},o)$ and its corresponding context distribution $\pi(\mathbf{c}|o)$ independently, since the sum over o from the mixture models $\pi(\mathbf{c})$, $\pi(\boldsymbol{\theta}|\mathbf{c})$ respectively appears in the log term. In the following sections we show, how we can overcome this limitation by introducing a lower bound inspired by variational inference [11]. As we consider each possible context as equally important, $p(\mathbf{c})$ is assumed uniformly distributed in a given interval in the following and thus, can be neglected in the objective.

## 3.2 Lower-Bound Decomposition for Component Distributions

In order to maximize the objective in Eq. (4) for each component $\pi(\boldsymbol{\theta}|\mathbf{c},o)$ individually, we can first extract the terms which only depend on $\pi(\boldsymbol{\theta}|\mathbf{c},o)$ for a specific o as

$$\max_{\pi(\boldsymbol{\theta}|\mathbf{c},o)} \mathbb{E}_{\pi(\mathbf{c}|o),\pi(\boldsymbol{\theta}|\mathbf{c},o)} \left[ R(\mathbf{c},\boldsymbol{\theta}) + \alpha \log \pi(o|\mathbf{c},\boldsymbol{\theta}) \right] + \alpha \mathbb{E}_{\pi(\mathbf{c}|o)} \left[ H \left[ \pi(\boldsymbol{\theta}|\mathbf{c},o) \right] \right]. \tag{5}$$

The responsibilities are still hindering us to optimize each component $\pi(\boldsymbol{\theta}|\mathbf{c},o)$ independently. However, similar to [11], we can obtain a tight lower-bound by introducing a variational distribution

$\tilde{\pi}(o|\mathbf{c}, \boldsymbol{\theta})$ and replacing the responsibilities in Eq. (5) with $\tilde{\pi}(o|\mathbf{c}, \boldsymbol{\theta})$. This variational distribution is fixed during the optimization and can be computed according to the last policy model allowing us to update each component independently. It is easy to show that after the update of the variational distribution $\tilde{\pi}(o|\mathbf{c}, \boldsymbol{\theta}) = \pi_{\text{old}}(o|\mathbf{c}, \boldsymbol{\theta})$, the introduced lower bound is tight. Please refer to the appendix. The resulting lower bound is a standard maximum entropy RL objective with an additional reward augmentation of $\alpha \log \tilde{\pi}(o|\mathbf{c}, \boldsymbol{\theta})$ and thus can be optimized with any suitable Policy Search algorithm. Here we use a maximum-entropy-adjusted version of contextual MORE [18].

### 3.3 Lower-Bound Decomposition for Context Distributions and Prior Weights

To update $\pi(\mathbf{c}|o)$ for a specific $o$, we extract the relevant terms from Obj. (4)

$$\max_{\pi(\mathbf{c}|o)} \mathbb{E}_{\pi(\mathbf{c}|o)} \left[ L_c(o, \mathbf{c}) + (\beta - \alpha) \log \tilde{\pi}(o|\mathbf{c}) \right] + \beta \mathrm{H}\left(\pi(\mathbf{c}|o)\right), \tag{6}$$

where $L_c(o, \mathbf{c}) = \mathbb{E}_{\pi(\boldsymbol{\theta}|\mathbf{c}, o)} \left[ \mathrm{R}(\mathbf{c}, \boldsymbol{\theta}) + \alpha \log \tilde{\pi}(o|\mathbf{c}, \boldsymbol{\theta}) \right] + \alpha \mathrm{H}\left[\pi(\boldsymbol{\theta}|\mathbf{c}, o)\right]$ corresponds to the expected augmented maximum entropy objective of component $\pi(\boldsymbol{\theta}|\mathbf{c}, o)$ in context $\mathbf{c}$ and $\tilde{\pi}(o|\mathbf{c}) = \pi_{\text{old}}(o|\mathbf{c})$ is a second variational distribution, which we introduced to be able to optimize for each $\pi(\mathbf{c}|o)$ individually. Similarly to the previous section, the objective given in Eq. (6) is a tight lower bound to the original objective where the responsibilities $\pi(o|\mathbf{c})$ are used instead of $\tilde{\pi}(o|\mathbf{c})$. We approximate the integral over $\boldsymbol{\theta}$ with a single sample, as we typically only have a single parameter sample per context available. Yet, as we still have the outer expectation $\mathbb{E}_{\pi(\mathbf{c}|o)}$ in Eq. (6) which we approximate by multiple context samples, the whole Monte-Carlo estimation of the expectations is still unbiased and with low variance. After the optimization step (Eq. 6), we obtain the optimal solution $\pi^*(\mathbf{c}|o)$ and tighten the bound by setting $\tilde{\pi}(o|\mathbf{c}) = \pi^*(o|\mathbf{c})$ and $\tilde{\pi}(o|\mathbf{c}, \boldsymbol{\theta}) = \pi^*(o|\mathbf{c}, \boldsymbol{\theta})$. Again, this lower bound corresponds to an augmented maximum entropy RL objective and thus we can be updated it with any suitable policy search method. Like [11], we use an adjusted version of MORE [17].

Finally, we can formulate the objective for updating the component weights $\pi(o)$, which resembles a lower bound of the original Objective (4) and corresponds to the highest hierarchy in our update scheme. The objective is given as

$$\max_{\pi(o)} \sum_o \pi(o) \left[ \mathbb{E}_{\pi(\mathbf{c}|o)} \left[ L_c(o, \mathbf{c}) + (\beta - \alpha) \log \tilde{\pi}(o|\mathbf{c}) \right] + \beta \mathrm{H}\left(\pi(\mathbf{c}|o)\right) \right] + \beta \mathrm{H}\left(\pi(o)\right), \tag{7}$$

which is a maximum entropy RL objective for categorical distributions. Here, we use REPS [19].

To summarize, we split the initial objective in Eq. (4) into different hierarchies, allowing us to optimize the different terms in our mixture model individually. Starting by first updating the components $\pi(\boldsymbol{\theta}|\mathbf{c}, o)$ using the maximization problem in Eq. (5), we can optimize for the component-wise context distributions $\pi(\mathbf{c}|o)$ with Obj. (6) after tightening the bound. The components $\pi(\boldsymbol{\theta}|\mathbf{c}, o)$ adjust the movement primitive parameters $\boldsymbol{\theta}$ given a context $\mathbf{c}$, while the component-wise context distributions $\pi(\mathbf{c}|o)$ ensure that the components only see context samples from a local context region. Finally, we update the weight distribution $\pi(o)$ using Eq. (7).

**Algorithmic Details and Addition of Components.** We initialize our algorithm with only one component and incrementally add components, and their corresponding context distributions, randomly. We fix all components except for the newly added one and optimize it for $K$ iterations to let it discover new solutions in yet undiscovered context regions. For updating $\pi(\boldsymbol{\theta}|\mathbf{c}, o)$ (Obj. (5)) we sample from the local context distribution $\pi(\mathbf{c}|o)$. By also updating $\pi(\mathbf{c}|o)$ according to Obj. (6), the components can adjust their curriculum and search for their favored context regions. After $K$ iterations we add a new component and repeat the procedure. Due to the augmented rewards, the new component will focus on undiscovered solutions and the local context distribution will cover uncovered areas of the context space. Note that such a simple adding procedure is only possible due to the mode-seeking properties of the I-projection, as the components do not need to average over multiple modes but can specialize on a local context region. We also fix the weights $\pi(o)$ to be uniformly distributed among all components during learning, since otherwise components which are already fully trained might dominate the optimization. By allowing to fine-tune all components every $H$ iterations, the previously added components can adjust to the newly added ones. After finishing adding components, we update the weights $\pi(o)$ at the end of our optimization procedure. The variables $K$ and the number of components added in total are task-dependent. We describe the algorithm in more detail in the Appendix B.

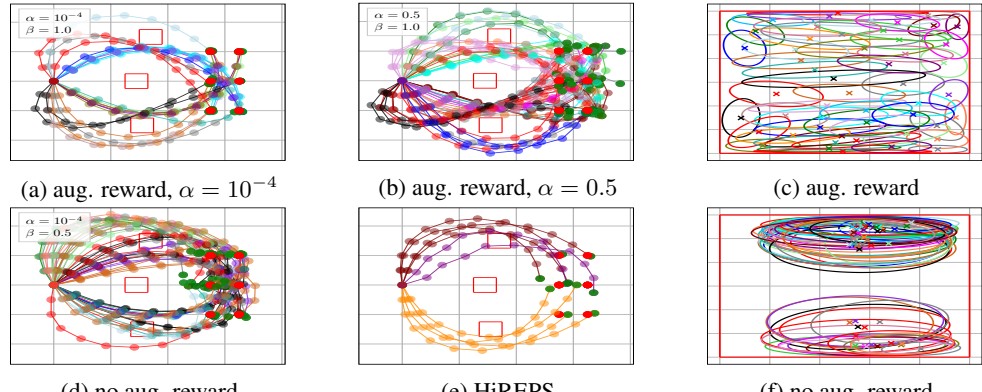

Figure 1: **Importance of the responsibilities in the augmented rewards and comparison to HiREPS.** A snapshot of the learned policies of the planar reaching task for a 2-dim context space (illustrated in (c) and (f)) by considering $\log \tilde{\pi}(o|\mathbf{c}, \boldsymbol{\theta})$, $\log \tilde{\pi}(o|\mathbf{c})$ ((a)+(b)+(c)) and neglecting the auxiliary distributions ((d)+(f)). In ((a)+(b)+(c)) we can learn more diverse solutions (a) + (b) and cover the whole 2d-context space (c) with different entropy bonuses, whereas the solutions in ((d)+(f)) show nearly the same, partially invalid solutions by going through the red rectangles (obstacles) (d) and are not able to cover the whole context space (f) leading to poor generalization performance. The solutions by HiREPS (e) indicate less versatility compared to the solutions (a) + (b). Note that each color indicates a different component and the red dots indicate the 6 chosen context vectors used for sampling.

## 4 Experiments

We start by investigating the importance of the different terms of the objectives derived in Section 3 and subsequently evaluate the versatility and precision of the learned skills on simulated robot beer pong and table tennis experiments. In Appendix C we report all hyperparameters and provide a comparison and analysis to a step-based policy search strategy (PPO [34]) in different scenarios.

### 4.1 Ablation Studies

We investigate the importance of the augmented rewards on a 10-link planar reaching task with two-dimensional context space. For this purpose, we update the component distributions $\pi(\boldsymbol{\theta}|\mathbf{c}, o)$ and the corresponding context distributions $\pi(\mathbf{c}|o)$ by optimizing the objectives in (5, 6) with i) considering the responsibilities (as given by our algorithm) and ii) by setting $\log \tilde{\pi}(o|\mathbf{c}, \boldsymbol{\theta})$ and $\log \tilde{\pi}(o|\mathbf{c})$ to zero. Furthermore, we compare the solutions found by our method to the solutions found by the SOTA method HiREPS.

**Versatility is Induced by the Augmented Rewards.** In the 10-link planar reaching task (see Figure 1a) the robot has to reach the red dots with its end-effector in a 2-dim context-space, while avoiding the rectangle-shaped obstacles. By adding 60 components, we have trained both versions (i and ii) over 15 seeds/trials and have chosen the best performing parameter constellations (Fig. 1a) for i), Fig. 1d) for ii)). We then picked the first model and sampled for each of the shown six contexts vectors (red dots) 100 samples and plotted the corresponding mean of each sampled component. The plots for i) (Fig. 1a) + Fig. 1b)) show versatile – several modes to the same context – and precise – small distance of end-effector (green dot) to goal (red dot), while avoiding obstacles– solutions while fully covering the context space (Fig. 1c). A trend of covering more modes with higher $\alpha$ can also be seen in (Fig. 1b)). For the case without the responsibilities ii) (Fig. 1d)), the solutions are not precise and invalid –reaching through the obstacles–. As Fig. 1f)) shows, the context distributions focus on easy-to-solve context regions (top and bottom part) and do not cover the full context space, leading to extrapolated solutions from components that are not trained for these contexts (red dots).

**Comparison to HiREPS.** We pick the best performing model among 15 seeds/trials after hyperparameter optimization. In Fig. 1e) the mean solutions of the sampled components –sampled in the same way as before – can be seen. HiREPS shows less diverse and qualitative solutions, where only 3 of the 60 components were chosen in total.

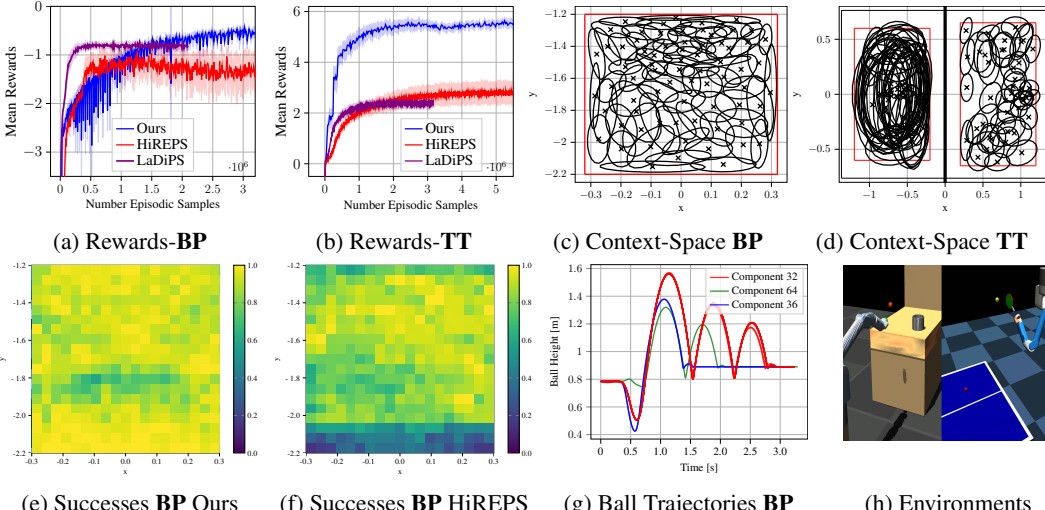

(a) Rewards-**BP**  (b) Rewards-**TT**  (c) Context-Space **BP**  (d) Context-Space **TT**

(e) Successes **BP** Ours  (f) Successes **BP** HiREPS  (g) Ball Trajectories **BP**  (h) Environments

Figure 2: **Beer Pong (BP) and Table Tennis (TT) Experiments.** In the **BP** experiment –left of (h)– the robot has to throw the red ball into a cup on the table. In the **TT** experiment –right of (h) – the robot hast to return the ball to the opponent's table side. The learned mixture of expert (MoE) models show high quality of the skills, reflected in the high reward values (a + b) and dense coverage of the context spaces, i.e., the 2-dim. position of the cup (c) for **BP** and the 4-dim desired outgoing landing and incoming landing ball positions (d) for **TT**. For **BP** we also compare the success rates (92% in average) of our approach (e) and HiREPS (75% in average) (f) throughout the context space, where we consider a trial successful if the ball goes into the cup. For **BP** versatile skills induce versatile ball trajectories for a given context (g). Versatile strikes for the **TT** experiment are shown in Fig. 3)

## 4.2 Simulated Robotic Experiments

We test and compare our algorithm on simulated Beer Pong and Table Tennis environments in Mu-JoCo [35], where we have chosen HiREPS [8] and LaDiPS [9] as baselines. Both methods are suitable baselines since they are contextual policy search algorithms and consider optimizing a mixture model. Throughout our experiments we choose probabilistic movement primitives (ProMP) [6] as policy representation, where depending on the weights $\boldsymbol{\theta}$ desired trajectories are generated and subsequently tracked with a PD-controller. In our experiments, given a context $\mathbf{c}$, the components $\pi(\boldsymbol{\theta}|\mathbf{c}, o)$ adjust the weight vector and the length of the trajectory, which are summarized in the vector $\boldsymbol{\theta}$. We consider non-markovian rewards, in which the reward depends on the history of state and actions. This type of reward function is not applicable to common step-based RL methods which build on markovian properties. As for analyzing the results of the experiments, we focus on the questions i) how does our algorithm perform compared to SOTA baselines, ii) are we able to cover the whole context space, and iii) are we able to learn versatile skills?

**Beer Pong.** The goal of the Barret WAM robot is to throw the red ball into the cup on the table. The 2-dim contexts resemble the position of the cup on the table. We incrementally add 70 components in our method, while HiREPS and LaDiPS directly start with 70 components. We have run all algorithms over 20 seeds/trials and report the performance in Fig. 2a)[1], where we plot the mean reward with two times the standard error. While HiREPS already converges after around half a million samples, our approach can quickly outperform it. Since LaDiPS uses intra-option learning, it outperforms our method in terms of sample efficiency. However, with the increasing number of components, we achieve a higher end-reward indicating that we can learn more qualitative solutions. Our algorithm allows to cover the whole context space with the learned component-wise distributions (see Fig. 2c)) resulting in a high end-reward. This performance is reflected in Fig. 2e), where we have divided the context space into a fine grid and sampled 100 times for each context a component from which we have executed the mean. We repeated that for all 20 different models and plot the mean success rate of throwing the ball into the cup. We did the same procedure for

---

[1]To reflect the model's true performance, the lastly added component was excluded from testing, since it is not fully trained yet and would not be chosen by the model if $\pi(o)$ would not be a uniform-distribution.

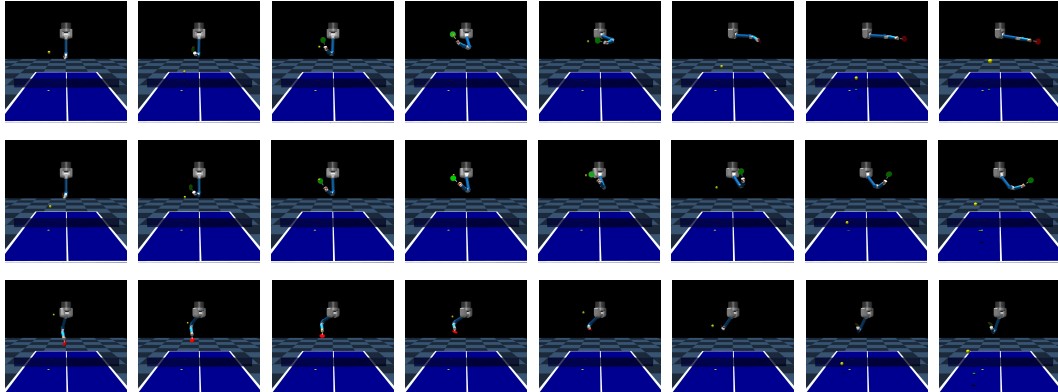

Figure 3: **Versatile Strikes for the Table Tennis (TT) Experiment** illustrated for a fixed context. The robot can hit the (yellow) ball in various ways, also indicated through the different colors of the racket sides. Note that the red and yellow dots on the table are markers for the serving and desired landing position respectively.

HiREPS in Fig. 2f). We can observe that HiREPS has much darker rectangles, showing that the success rate in these context regions is low. Although versatility is encouraged in the joint space of the robot, different joint trajectories often yield different ball trajectories. Given one context, in Fig. 2g) we show the z-coordinates of the ball trajectories over time resulting from sampling 20 times from the MoE model. Each component leads to a different number of "ball-jumps". In Appendix C.2.1 we qualitatively show that we can learn more versatile solutions than LaDiPS and report that we can achieve a much higher expected mixture entropy (Fig. 4), which indicates that our solutions are versatile.

**Table Tennis.** The task of the robot is to return different incoming balls to desired targets on the opponent's table side in different ways. We consider a four dimensional context space, including ball's initial serve position and ball's target landing position (right and left half of table Fig. 2d)). For both parts of the context, we fix the z position and vary the x and y coordinates. We incrementally add 50 components in our method, while HiREPS and LaDiPS directly start with 50 components. We have run all algorithms over 20 seeds/trials and report the performance in Fig. 2b)[1], in which the mean reward with two times the standard error is plotted. The component-wise context distributions are spread among the context space (Fig. 2d)) and allow each component to locally specialize on a context region. This high coverage of the context space allows for a high reward achievement, outperforming HiREPS and LaDiPS. In Fig. 3), we show three different striking skills sampled from our trained MoE model, for a fixed context, i.e., fixed serving and desired ball position. The first two skills use the green side of the racket to hit the ball (forehand), while the third skill uses the red side of the racket (backhand) to hit the ball. In contrast to the second strike, the first one performs a smash-like strike and ends it with the red side of the racket pointing to the camera.

## 5   Conclusion

We proposed a new objective for learning contextual and versatile Mixtures of Experts (MoE) models. We based our objective on a maximum entropy formulation to increase the versatility of the solutions and introduced a curriculum to allow the components to specialize. Our formulation also allows for easy online adaptation of the model complexity during training. We conducted an ablation to show the importance of the individual parts of our objective. Further, we showed that our method learns precise and versatile solutions and outperforms the baseline on sophisticated simulated robotic tasks. This work aims at presenting a mathematically well-founded method and demonstrates its general feasibility on various challenging tasks. Currently, the major drawback of our approach is sample efficiency, as we do not share experience between the components. We intend to address this issue in future work, e.g., by intra-option learning. Another direction for future research is extending the approach to more complex models, such as non-linear mixtures of experts. We expect to need fewer components to cover the whole context space with more complex component model representations.

**Acknowledgments**

Calculations for this research were conducted on the high performance computer of the state of Baden-Württemberg.

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
