# OpenReview forum: "Specializing Versatile Skill Libraries using Local Mixture of Experts"
_robot-learning.org/CoRL/2021/Conference — CoRL2021 Poster_

### Official Review · Reviewer_vGwW · 2021-07-23

**Originality:** Good
**Technical Quality:** Very Good
**Clarity Of Presentation:** Good
**Impact:** 4

**Recommendation:**

Weak Accept: I recommend accepting the paper, but will not argue for my recommendation if the majority of other reviewers have a different opinion.

**Summary:**

This paper proposes a novel objective function for learning a Mixture-of-Experts (MoE) model to solve robot motion tasks. The objective function uses maximum entropy to encourage diversity in a collection of learned motion primitives, which are called "components." An interesting aspect of this paper, in comparison to existing work, is that the motion primitives are added and learned sequentially, with the objective function encouraging each one to differentiate from the others, thereby covering the entire space of "contexts" (i.e., different variations of the same task). The method outperforms the existing HiREPS method in simulation on difficult tasks, justifying the underlying math that went into creating the objective function.

Update 1 Sept: I am more confident in my "accept" recommendation now. I'll leave it as a "weak accept" because, per the authors' comments, it seems I'm not super knowledgeable about this area. But, I think the general consensus on this paper is an "accept" with the understanding that it's interesting and good, but maybe not mind-blowing or completely revolutionary (but still good!).

**Issues:**

Abstract
- It would be helpful to note any comparison methods in the abstract.

Introduction / Related Work
- The claim that "most of these algorithms... usually assume a Gaussian policy" needs some citations to back it up; I think [2] in the paper is the relevant one?
- For the proposed method, depending on the dimension of the context space, it would take an exponential number of local regions to effectively cover it. How can we avoid this blowup? I guess removing suboptimal solutions helps out.
- I would appreciate a sentence that clarifies the "ad hoc constraint" for HiREPS and LaDiPS (i.e., just add more details explaining the constraint)

Specializing Versatile Mixture of Expert Models
- The terms R and H in the maximum entropy aren't defined (one can figure out that R is reward and H is entropy, but it would be good to state it explicitly)
- It would be helpful to number all equations for the sake of reviewers :)
- On line 149, the text refers to "log terms of the entropy and the KL" but these log terms haven't appeared explicitly yet (i.e., the entropy and KL terms should be expanded before (3) so it's clearer for the reader)
- Why is it reasonable to approximate the integral over $\theta$ with a single sample? I feel like this statement needs further justification
- How does one know when a component is sufficiently trained before adding a new one? How does one know that enough components have been added? While these questions may be answered in Appx. B, it would be helpful to at least briefly answer them in the main text.

Experiments
- In Fig. 1, I presume that the dotted lines are the 10-link robot. What are the red boxes? I guess obstacles? (these questions are answered in the main text around line 228, but should be explained in the figure caption as well)
- I'm confused by how the beer pong task shows versatility; I think it's best shown by the way that the robot bounces the ball multiple times. But it feels like true versatility would be for a single MoE model that learns _both_ beer pong and table tennis, or something really wild like that. I realize that I'm asking for a lot, but I think that the paper should have a much more precise statement about what "versatility" means (i.e., it learns many variations of a single task, as opposed to learning multiple tasks, and variations, with a single model)
- To be clear, I do think the table tennis task shows versatility :)

**Reviewer Expertise:**

Fair: Some knowledge of the area

**Strengths And Weaknesses:**

STRENGTHS
- I love the idea of local context distributions that adapt and grow; this feels a lot like human learning, and it makes sense that the approach would be very versatile.
- I really appreciate the clear statement of strengths/weaknesses of the related methods
- It is clear the authors put effort into stepping the reader through their thought process
- The comparison to HiREPS seems appropriate and well-justified

WEAKNESSES
- The word "versatility" in the title is never formally defined, so the assessment of versatility seems entirely qualitative, but it would ideally be quantitative
- I feel that the word "context" is a bit misleading when considering what is meant by a context in this paper; here, contexts are limited to a continuum of task properties (e.g., goal position or friction parameters). I would generally imagine contexts to be a much higher-level abstraction, such as "kitchen" versus "living room." But, perhaps this is just a part of the MoE vocabulary with which I'm not familiar?
- Some justification for claims in the paper are in the appendices, but I think it would be better if they could be brought into the paper; I've made note of them below

**Summary Of Recommendation:**

I am putting "weak accept" because I am not super familiar with mixture-of-experts or skill library learning. I think this paper is pretty good, and just needs some clarifications to be acceptable. But, if it's already been done or other more knowledgeable reviewers have serious problems with it, then I understand rejecting it.

---

> ### Author Response · Authors · 2021-08-30
> **Reply to Reviewer vGwW**
>
> * *"The word "versatility" in the title is never formally ..."*
>     * We agree that a more formal definition of versatility would be useful for the community, however, also other approaches in the literature are lacking such definitions as versatility is hard to define as well as hard to measure quantitatively without relying on task-specific prior knowledge. We are not aware of any useful measure used in other approaches that strive for versatility. However, we gave a more direct explanation of what we mean with versatility (i.e. to discover multiple solutions) in the introduction.
>
> * *"I feel that the word "context" is a bit misleading when considering ..."*
>     * We followed the definition of contextual policy search as described in the works [6, 14, 15, 16, 17], where the context is defined as we did in the paper.
>
> * *"It would be helpful to note any comparison methods in the abstract."*
>     * We thank the reviewer. We have added a sentence to the abstract.
>
> * *"The claim that "most of these..."*
>     * We thank the reviewer. We have added several citations
>
> * *"For the proposed method, depending on the dimension ..."*
>     * The number of required local regions naturally depends on the dimensionality of the context space, however, it also adapts to the complexity of the problem. I.e., if the adaption for certain context variables is almost linear, the valid local regions are bigger, and therefore fewer regions are needed. Hence, the "exponential blowup" is more dependent on the complexity of the task instead of the dimensionality of the context space. We agree that for high dimensional context spaces with highly non-linear adaptation relationships this exponential increase of local regions is an issue, however, such problems constitute a challenge for other methods, such as neural networks, as well as it is hard to find good generalization strategies for such problems.
>
> * *"I would appreciate a sentence ..."*
>     * We have added two sentences explaining the additional constraints for HiREPS and LaDiPS
>
> * *"The terms R and H in the maximum entropy ..."*
>     * We have mentioned in the paper, that the terms R and H stand for the reward and entropy respectively. Thanks for pointing it out.
>
> * *"It would be helpful to number all ..."*
>     * In order to provide improved discussion we followed the reviewer's suggestion and numbered all equations
>
> * *"On line 149, the text refers to "log terms of the entropy and the KL" but ..."*
>     * To avoid confusion, we have included the definition (including the logs) in Section 3.1
>
> * *"Why is it reasonable to approximate the integral over ..."*
>     * We assume to only have one parameter sample for each context as it is hard to exactly repeat the same context twice. This is a common strategy in contextual episodic reinforcement learning [3]. Note that a single sample estimate of the integral is an unbiased monte-carlo estimator (with high variance). As the expectation in Eq 6 (updated document) is approximated with multiple monte-carlo samples from the contexts $\boldsymbol c$, the variance of the single-samples estimates for $L_c$ (one $\boldsymbol \theta$ per $\boldsymbol c$) is reduced while the whole monte-carlo approximation still stays unbiased. We clarified that in the text (Section 3.3).
>
> * *"How does one know when a component is sufficiently..."*
>     * These parameters are task-dependent and currently need to be chosen by the user. However, the number of iterations needed for convergence for one component can usually be found out by only adding one component and letting it run for x iterations. Convergence is quite easily detectable but is right now chosen manually. The number of components added in total is depending on the desired performance. As soon as the algorithm converges, we can stop adding components. We have added a small description at the end of Section 3.3 to clarify these points. We thank the reviewer for pointing this out.
>
> * *"In Fig. 1, I presume that the dotted lines are the 10-link robot. What are the red ..."*
>     * We thank the author for the information. We have extended the caption of the Figure and mentioned that the red rectangles are the obstacles

---

> > ### Author Response · Authors · 2021-08-30
> > **Reply to Reviewer vGwW continued**
> >
> > * *"I'm confused by how the beer pong task..."*
> >     * Yes, as described in the Beer Pong experiment, versatility in the Beer Pong task is learned in the joint space, as the ProMPs influence the joint trajectory of the robot. However, different joint trajectories also lead to different ball trajectories resulting in different 'ball jumps' before landing in the cup. We also tried to emphasize this in the video of the supplementary material. We consider versatility in the solution to a given context of the main task. e.g. we want to learn different 'throwing solutions' in the Beer Pong task for the same context (position of the cup). We want to achieve this for all possible contexts of the the continuous context space, which is the whole table in the Beer Pong task. We tried to make the definition more clear in the Introduction.

---

> > > ### Comment · Reviewer_vGwW · 2021-09-01
> > > **thanks for response**
> > >
> > > I apologize for not writing this sooner -- I appreciate that the authors addressed all of my concerns, and feel more confident in my "accept" recommendation now :)

---

### Official Review · Reviewer_xPhY · 2021-07-24

**Originality:** Good
**Technical Quality:** Good
**Clarity Of Presentation:** Fair
**Impact:** 3

**Recommendation:**

Weak Accept: I recommend accepting the paper, but will not argue for my recommendation if the majority of other reviewers have a different opinion.

**Summary:**

Learn a set of motion primitives that can achieve various parameterized behaviors. Then learn an expert agent which chooses the parameters of the behavior from the task definition. The expert policy is optimized for selection of motion primitive parameters using maximum entropy reinforcement learning, which regularizes with the entropy of the policy given context. An additional regularization on the distribution of the context vector pushes the KL distributional distance towards uniform. This learning can be broken into three distributions, the distributions over the component weights based on context and primitives, based on context, and the raw distribution. These can be used by Bayes identities to determine the primitive and expert policies.

**Issues:**

I believe the work to be interesting and have good results. However, it is difficult to make that assessment with confidence because the paper itself is hard to parse. Some changes to the clarity of the derivation and the intuitive definition of terms, grounded in the experiments, would greatly improve the paper.

**Reviewer Expertise:**

Good: General knowledge of the area

**Strengths And Weaknesses:**

Strength:

Proposes an effective derivation of an entropy based distribution of behavior

A strong collection of experiments that visualize an otherwise complex description

The experiments are interesting and provide encouraging results on robotics tasks. A physical robot would have been nice but not necessary

Weaknesses:

While assuming that p(c) is uniform simplifies the derivation, it is not clear whether this assumption has detrimental effects on the final policies

The clarity of the derivation between equations 2 and 3 is somewhat confusing. The identity only becomes clear when looking at the appendix, where the order of terms and the values is made clear. However, even after the derivation is identified, it is hard to follow the flow of the equation because there is little intuitive description of the six terms in the equation. The work provides little to no intuitive meaning for terms, as the subsequent sections mostly focus on describing the auxiliary distributions used to optimize them.

This work introduces a large number of terms, including describing mixture of experts, component weights, context vectors and primitive parameters. However, it is not clear what the intuitive interpretation of these terms is, as they are only described vaguely in the introduction. Even in the experiments, it is still not abundantly clear what the intuitive meaning of these terms are without familiarity with the methods used (ProMP). Considering that the work handles a good amount of derivation, it would be useful to have a more in-depth description of the intuitive meanings, especially since final terms are not particularly clear.

Some discussion in related work of mutual information based skill learning methods, such as Explore, Discover and Learn: Unsupervised Discovery of State-Covering Skills (Campos et. al.), or other work in the hierarchical reinforcement learning literature would be useful, since this is a work that uses a different jargon but aims to achieve very much the same set of hierarchical decision-making.
One minor limitation is that though the derivation is general, the results rely heavily on ProMPs for the primitives, which means that the executed behaviors tend to be particular smooth motions, rather than adaptable behaviors which might be possible with a less dynamically based policy class such as a neural policy.


**Summary Of Recommendation:**

Most of the score of weak accept is given because it was difficult to parse this work and capture the intuitive meanings. The method and derivation seem sound and the novelty is good, but again, because it is somewhat difficult to follow the paper, it is difficult to have high confidence in the result.

---

> ### Author Response · Authors · 2021-08-30
> **Reply to Reviewer xPhY**
>
> We want to thank the reviewer and want to answer his concerns in the following
> * *"While assuming that p(c) is uniform simplifies the derivation, it is ..."*
>     * We consider a Mixture of Gaussians $pi(\boldsymbol c)$ for the context distribution, which is a powerful representation. Thus, with a sufficiently large number of active components, we can approximate any complex distribution. This is encouraged by the KL-term in objective (3). Note that our derivation does not depend on the density $p(\boldsymbol c)$ being uniform. As long as $p(\boldsymbol c)$ is known the derivation and the algorithm can be extended straightforwardly to non-uniform distributions.
>
> * *"The clarity of the derivation between equations 2 and 3 is somewhat confusing. The identity..."*
>     * We thank the reviewer for the valuable feedback. We have extended Section 3.1, 3.2, 3.3 and added more intuitive explanations of the individual terms. Furthermore, we rewrote Equation (4) (previously Equation (3)).
>     * In general, the reformulation from Objective (3) to Objective (4) allows to obtain key elements for our algorithm: the maximum entropy objective for each component $\pi(\boldsymbol \theta|\boldsymbol c,o)$ augmented with the log responsibilities and the maximum entropy objective for the component-wise context distributions $\pi(\boldsymbol c|o)$ also augmented with the log responsibilities. With some further modifications, Objective (4) allows to update each component independently and the combination of both, i.e. maximum entropy and log responsibility augmented reward, leads to the desired behavior, which is demonstrated in Section 4.
>
> * *"This work introduces a large number of terms, including ..."*
>     * We thank the reviewer for pointing out the unclear descriptions. We have added a more detailed description of the individual terms of our mixture model in Section 3 and tried to give a more detailed intuition for the terms emerging in the objective (now equation 4) in Section 3.1 and Section 3.3. Furthermore, we tried to give a better intuitive meaning of the individual terms in relation to ProMPs in Section 4.2
>     * In general, given a context $\boldsymbol c$ (e.g. 2d cup position in the Beer Pong task), the chosen component samples a weight vector $\boldsymbol \theta$, which is subsequently used to generate the desired trajectory by multiplying the weights with basis-functions. Those basis functions are determined per joint and are predefined (principle of ProMPs). By using a PD-controller, torques are generated and fed into the system. In the case of a position-controlled robot, in each time step, the desired position of the trajectory is fed into the system.
>
> * *"Some discussion in related work of mutual information based ..."*
>     * We thank the author for this informative suggestion. We have added the work from Campos et.al. to our related work section. Yet, we want to mention that most of the step-based approaches are based on the mutual information objective, but are reformulated as a maximum entropy objective. Our formulation is more direct as we directly start with a maximum entropy objective. To make this clear, we have reformulated the corresponding part of the related work. Furthermore, we included the hierarchical RL approach of the options-framework into our related work section.
>     * We agree with the reviewer. The experiments consider a contextual episodic policy search setting, where the skills are encoded as movement primitives. We want to emphasize that ProMPs are the choice we did in this paper but other movement primitives such as DMPs can also be used. We agree that once the skill to be executed is chosen, there is no adaptable behavior until the next skill. However, we also want to emphasize the advantages of this kind of skill parametrization: First, the episodic setting enables the usage of non-markovian rewards, which are usually much easier to design. Second, these approaches do exploration in the parameter space rather than in the raw action space, which is in the setting of more sparse rewards often beneficial. To showcase this, we have designed different settings on the 5-Link Reacher task (a more complex version of the 2-Link environment from OpenAi Gym). The analysis can be found in Appendix C4.

---

> > ### Comment · Reviewer_xPhY · 2021-09-08
> > **Response to Authors**
> >
> > I appreciate the well-considered responses to concerns over clarity and giving a better understanding of the work in general. The rewrites to the unclear sections improved the clarity and helped with gaining an intuitive understanding of the work, which was missing in the original.

---

### Official Review · Reviewer_as3F · 2021-07-24

**Originality:** Very Good
**Technical Quality:** Good
**Clarity Of Presentation:** Fair
**Impact:** 4

**Recommendation:**

Weak Accept: I recommend accepting the paper, but will not argue for my recommendation if the majority of other reviewers have a different opinion.

**Summary:**

The paper proposed a learning algorithm for obtaining a mixture of control policies that solves tasks with contexts in different ways. The main contribution of the paper is an objective that allows different expert policies (components) to be learned independently, thus enabling incremental learning of different components. They demonstrate that the proposed method can achieve good coverage on the context space of the task and good performance of the tasks.


**Issues:**

- A notation issue: \pi is used everywhere in the paper, what does it stand for in this case? Initially I interpreted it as the control policy, but later realized it is more like a probability distribution, which is a bit confusing.
- The identities in line 151 are not very clear: should there be a summation over o in it? Otherwise what should o be given that it doesn’t appear on the left-hand side?
- The result in the video shows that the method can learn to solve a particular task with different styles. Are those styles from different components, or do they emerge from the max-entropy objective under the same component?
- Add discussion regarding if the proposed method can handle tasks with high-dimensional context space: are all the terms in the algorithm easily scalable to high-dimensional spaces?
- Also, how would the method work if different contexts correspond to tasks with very different difficulties? Would policies attempting to solve those tasks get constantly pruned due to being suboptimal?
- Ideally, add some comparisons to LaDiPS [8].


**Reviewer Expertise:**

Good: General knowledge of the area

**Strengths And Weaknesses:**

+ The idea of incremental learning of components that covers different parts of the context space is interesting.
+ The experiment result shows interesting behaviors learned by the proposed method. The comparison to prior methods in general supports the idea in this work.

- The writing is not very clear in general, making the paper not very easy to understand. More details in issues below.
- It’s not clear if the proposed method can handle tasks with high-dimensional context space, or with varying difficulties for different contexts.
- Though the paper provided comparison to the HiREPS algorithm and showed superior performance, it seems that LaDiPS [8] is a better baseline in this case given that it shares the same policy search algorithm as the presented work. It’ll better support the value of the proposed incremental training scheme.


**Summary Of Recommendation:**

The paper presents a concrete learning algorithm and demonstrates good performance on a suite of difficult problems. However, the exposition of the paper could be improved and more discussions/comparisons would make the paper even stronger.

---

> ### Author Response · Authors · 2021-08-30
> **Reply to Reviewer as3F**
>
> We want to thank the reviewer and want to answer his concerns in the following
> * *"A notation issue: \pi is used everywhere in the paper, what..."*
>     * We use $\pi$ as a symbol for all adaptable distributions through the optimization process. In general, every $\pi$ is a probability distribution, also the control policy. In episodic search, we usually consider a stochastic search distribution sometimes also referred to as policy, over the parameters of a motion primitive.
>     * We have added a more detailed explanation at the beginning of section 3 to clarify this confusion.
>
> * *"The identities in line 151 are not very clear: should there ..."*
>     * The identities in line 151 (in newer version line 163 and 164) arise through Bayes'
> $$
> \pi(\boldsymbol \theta|\boldsymbol c,o)\pi(o|\boldsymbol c)=\pi(o|\boldsymbol c,\boldsymbol\theta)\pi(\boldsymbol\theta|\boldsymbol c) \rightarrow \pi(\boldsymbol\theta|\boldsymbol c)=\frac{\pi(\boldsymbol\theta|\boldsymbol c,o)\pi(o|\boldsymbol c)}{\pi(o|\boldsymbol c,\boldsymbol\theta)}
> \rightarrow \log\pi(\boldsymbol\theta|\boldsymbol c)=\log\frac{\pi(\boldsymbol\theta|\boldsymbol c,o)\pi(o|\boldsymbol c)}{\pi(o|\boldsymbol c,\boldsymbol\theta)}
> $$
> $$
> \pi(\boldsymbol c|o)\pi(o)=\pi(o|\boldsymbol c)\pi(\boldsymbol c) \rightarrow \pi(\boldsymbol c)=\frac{\pi(\boldsymbol c|o)\pi(o)}{\pi(o|\boldsymbol c)}\rightarrow \log\pi(\boldsymbol c)=\log\frac{\pi(\boldsymbol c|o)\pi(o)}{\pi(o|\boldsymbol c)}
> $$
> They are valid for all $o$.
> * *"The result in the video shows that the method can learn ..."*
>     * The different styles emerge from different components. Thanks for pointing that out. We will make that more clear for the camera-ready presentation
> * *"Add discussion regarding if the proposed method ..."*
>     * In general, the decomposed objectives do not make any assumption on the parametrization of the components (i.e. linear model, non-linear model), nor does it make any assumption on the method used to optimize the individual objectives. Therefore, this question can be answered in two ways: 1. Does the chosen optimizer allow for effectively updating each term given a higher-dimensional context space and 2. How many components do we need to fully cover the context space?
>     * 1. This point is rather independent of the decomposition proposed, since any optimizer can be considered. We, therefore, did not work this out more in detail in the paper.  However, still, we want to mention some details here. We used MORE to update the context distributions $\pi(\boldsymbol c|o)$ and Contextual MORE to update $\pi(\boldsymbol \theta|\boldsymbol c, o)$. We want to emphasize, that any other method can be used. The crucial part here is Contextual MORE, since a quadratic surrogate function for the joint Reward $R(\boldsymbol c, \boldsymbol \theta)$ is fitted. However, Contextual MORE scales linearly with the dimension of the contexts and thus was not a problem for our experiments. Nevertheless, for very high-dimensional problems the authors from [17, 18] have proposed different dimensionality reduction techniques.
>     * 2. A linear component model can not generalize to a bigger context region as well as a suitable non-linear model would. Thus, with linear models, we expect that we need more components to cover the whole context space in total. However, since we optimize one component at a time and then add a new component, the algorithm scales well to a large number of components. Nevertheless,  we plan to extend our models to non-linear models with Neural Nets. We consider this as an important point and have extended the Conclusion, where we emphasize the relation to non-linear models and the number of needed components to properly fill the context space.
>
> * *"Also, how would the method work if different contexts..."*
>     * We believe the reviewer meant components with 'policies' and would like to answer the question accordingly.
>     * This effect is considered in the ablation studies: For the case without the augmented reward, i. e. responsibilities, the component-wise context distributions concentrate on context regions that are easier to solve (see Fig. 1 e)). When considering the responsibilities as augmented rewards, this effect can not be observed (see Fig 1b)), since the firstly added components will already cover those easy-to-solve context regions. Components that are subsequently added will concentrate on context regions that are not covered yet due to the log responsibilities, even though these contexts are harder to solve. Since the weight distribution $\pi(o)$ is updated based on the hierarchical rewards (i.e. including the responsibilities), even the hard-to-solve contexts will have high weight and thus are not pruned. We have extended the corresponding section in the ablation studies to emphasize this point. Thanks for pointing that out

---

> > ### Author Response · Authors · 2021-08-30
> > **Reply to Reviewer as3F continued**
> >
> > * *"Ideally, add some comparisons to LaDiPS [8]."*
> >     * LaDiPS follows a different strategy than HiREPS and our method. It lower bounds the gating entropy and thus the model is forced to have a certain number of active components. In contrast, HiREPS and our method autonomously choose the number of active components allowing to adjust the model by themselves. We, therefore, believe that comparing to HiREPS is fairer and reflects the performance of the method. However, we have implemented LaDiPS and run it on both environments (Beer Pong and Table Tennis) (see Fig. 2a) +2b)). Since LaDiPS benefits of experience sharing and directly starts with 70 components, it is able to outperform our algorithm in terms of sample efficiency on the Beer Pong task. However, with an increasing number of components, we are able to outperform LaDiPS in terms of end-reward and thus learn more qualitative solutions. In Appendix C.2.1 we also show that we are able to learn more versatile solutions than LaDiPS. On the Table Tennis environment (Fig. 2b)), we are able to outperform LaDiPS, which is performing similarly to HiREPS.

---

> > > ### Comment · Reviewer_as3F · 2021-09-02
> > > **Thank you for the response**
> > >
> > > The response provided by the author addressed most of my concerns and the revised text notably improved over the initial version. Thus I believe it is a concrete work that can make good contribution to the CoRL community.

---

### Official Review · Reviewer_Kbdi · 2021-07-26

**Originality:** Good
**Technical Quality:** Good
**Clarity Of Presentation:** Fair
**Impact:** 3

**Recommendation:**

Weak Reject: I recommend rejecting the paper, but will not argue for my recommendation if the majority of other reviewers have a different opinion.

**Summary:**

The paper goes after two issues
1) How to learn to handle context (parameter) changes in the learning tasks, such as a variety of goals or friction values.
2) How to learn to execute different strategies to do the same thing (use a forehand or a backhand swing).
The suggests a mixture of experts approach. A more useful terminology emphasizes that policies are composed of classifiers that
select "simple" policies depending on the context and desired strategy. I would suggest the term "classifier-based policies" or
"selection-based policies". Mixture of experts often don't have a classifier and use weighted blends of the experts' outputs, which
is a very different thing.

Key ideas include:
1) Develop a curriculum to support learning the classifier and the component policies. This helps avoid interference during learning, as
well as getting good coverage.
2) Control the scope of the component policies by optimizing the worst case for each component.
3) Optimize the classifier and the component policies simultaneously (vs. in some EM-like alternation).
4) Iteratively add components to improve coverage.

**Issues:**

See the issues discussed above.

Clearly explain (so someone could actually write code) a simple example like the multi-joint reaching.

Implement behaviors on a real robot.

In the wonderful world of open review, all the submitted papers are
available to the authors. There are at least 3 "generate and test"
papers:
190 Specializing Versatile Skill Libraries using Local Mixture of Experts
98 Data-efficient learning of object-centric grasp preferences
89 Hierarchical Policies for Cluttered-Scene Grasping with Latent Plans
You should take advantage of this opportunity to put your work into
a larger context (and take a look at the reviews these papers got.)


**Reviewer Expertise:**

Good: General knowledge of the area

**Strengths And Weaknesses:**

Strengths:

I am very happy to see a paper on learning multiple "strategies" to do the same thing, as I think this is a very important issue
for skill learning.

Weaknesses:

It is easy to do this kind of stuff in simulation. Not so easy on real robots. Not having an implementation of something on a real robot
is a big weakness.

The paper promises "two challenging simulated robot skill learning tasks". Implementations of both of these tasks have been done
on real robots
in the past. Ball throwing is not that hard (see 1988 paper
https://www.researchgate.net/publication/37596892_Model-Based_Robot_Learning). Robot ping pong (table tennis) is challenging,
but perhaps the major part of the challenge is perception rather than action. Lots of people have implemented robot ping
pong players. Search for "robot ping pong table tennis players" on Youtube. I would back off "challenging" unless I was doing some
task that really hasn't been done many times before on a real robot. Or demonstrate real robot performance on a larger range of tasks
with appropriate generalization across tasks: Simultaneously learn: tennis, squash, badminton, paddle ball, racquetball, table tennis, soft tennis, platform tennis, beach tennis, smash ball, hand tennis, basque pelota, ...

Did the paper show suitable generalization across contexts? Was interference avoided? How was this evaluated?
At the end it says "Currently, the major drawback of
our approach is sample efficiency, as we do not share experience between the components."
What about between contexts when the same option is selected? When a different option is selected?

The motivation and notation for Equation 1 is not clear.
Given c, the option o is selected by the agent. This function is known, so why not use it directly instead of using Bayes rule to replace it.
pi(theta|c) = sum_o pi(theta|c,o) p(o|c)
Bayes rule applies to probabilities such as prob(o|c)prob(c) = prob(c|o)prob(o).
To me, the policy pi(o|c) is clearly prob(o|c).
The mention of Bayes' Rule in the derivation of equation 1 implies
pi(c) is not meaningful, prob(c) is, and has nothing to do with the agent. So how is pi(c) not equal to prob(c)?
pi(c|o) and pi(o) are not meaningful either, but prob(c|o) and prob(o) are and can be computed from pi(o|c) = prob(o|c), which is under
the control of the agent.
Clearly I am confused.

I don't understand why an entropy bonus is useful in the case where c has nearly irrelevant components. Won't this favor adding  unnecessary theta variance or irrelevant component policies? A useful regularization in reinforcement learning is to penalize complexity.
Can you explain why the entropy bonus is useful, and what practical effect it has.

I don't see why pi(c|o) or pi(o) (really the probabilities, that are only under indirect control by selecting pi(o|c)) are useful.
The difference between pi(c) and probability(c) is not clear. I shouldn't have to read Kink et al. [2] to understand it.
pi(c) is presumably the probability of c, and is not under control of the agent, so the notation should be p(c) or prob(c).
What is the meaning of pi(c|o)? The option selected has no effect on the actual context. Perhaps you should introduce a context
estimate hat(c) = sum_o pi(hat(c)|o)pi(o)

The distinction between trial-based and time-step-based learning algorithms was not clear. Perhaps "step-based" is too abbreviated in
"Related Step-Based Approaches. In the step-based setting, the policy does not learn a function
from contexts to parameters of an episodic controller but directly maps from system states to actions."
I might try to make the distinction clearer by emphasizing the difference between training policies
a) by adjusting a function from a context or goal to a small number of policy parameters at the end of trials (trial-based? episodic?)
vs.
b) by adjusting a function from states to actions at each time step of execution (time-step-based?)

For parametric component policies, I don't think there is a fundamental distinction between episodic and time-step-based approaches, especially
when there are no episodes (life-long learning).

It might be useful to make a connection to "options" as described by Sutton.

Given the tasks chosen are deterministic (at least when done with real robots),
it may also be useful to consider the deterministic model-based case, where selector() and pi_option() are deterministic.
pi(action|state) = Sum( selector(state)*pi_option(state) )
If the selector is one for one option at a time, this selects an option.
Anything else blends options. Time-step-based policies (options) can
be learned in the same way as episodic policies. This leads to much
simpler gradient learning rules than the stochastic case.
www.cs.cmu.edu/~cga/papers/acc2012-tr.pdf

How good are the lower bounds, and how much does it matter?

The paper promises "unprecedented accuracy and versatility," How is this demonstrated? Too much hype?







**Summary Of Recommendation:**

How to represent multiple strategies to do the same task (same initial conditions and goal) is an important question.
If multiple strategies were demonstrated on a real robot, that would be very interesting.

---

> ### Author Response · Authors · 2021-08-30
> **Reply to Reviewer Kbdi**
>
> We want to thank the reviewer and want to answer his concerns in the following
> * *"The paper promises ”two challenging simulated robot skill learning tasks”.  Implementations of both of these tasks have been done on real robots in the past.  Ball throwing is not  ..."*
>     * We agree that these kinds of tasks have been conducted on the real robot already. Still, there are remarkable differences to our set up, which is why we consider our provided environments challenging: i) we learn and discover skills for a 7DoF robot, which require precise and fast movements, from scratch (unlike previous works, where often either the underlying dynamics are known, or demonstrations are provided to the agent (at least for the ping pong task)), ii) we consider a high dimensional continuous context space, iii) we emphasize on versatility and hence learn multiple solutions. We are not aware of any other approach that is able to discover such versatile skills on tasks of similar complexity, even though our results are only achieved in simulation.
>
> * *"Did the paper show suitable generalization across contexts? Was interference..."*
>     * Unfortunately it is not fully clear to us what is meant by 'interference'. We believe that the reviewer meant overlapping regions in the context and context-parameter space. Following this definition of 'interference', we want to answer the reviewer's question in the following.
>
>     * The performance and generalization across the contexts are reflected in the reward curves, where we have sampled test-contexts from the environment's context distribution $p(\boldsymbol{c})$ (i.e. not from the learned context distributions $\pi(\boldsymbol{c})$) and then executed the skills sampled from the mixture model. The high resulting reward using the context distribution $p(\boldsymbol c)$ for evaluation implies that the models can cover the full context space of the environment, confirming generalization to unseen contexts (see (Fig. 1c), Fig. 2c) + Fig. 2d)). The full interference of the context distributions is avoided through the augmented rewards, i.e. considering the responsibilities in the updates (see Ablation studies in Section 4.1). However, to a certain amount, overlapping regions occur, which is due to the maximum entropy objective and which is also crucial to learn versatile skills. This effect is however not harmful as shown by the reward curves.
>
>     * Currently, we do not mix the samples each component has seen. Every component has its own small replay buffer. Consequently, no information between individual components is shared at all.  Yet, as the focus of this paper is on discovering a versatile set of skills rather than on sample efficiency, we consider extending the method to intra-option learning as future work
>
> * *"The motivation and notation for Equation 1 is not clear. Given c, the option..."*
>     * We thank the reviewer for the comment and would like to clarify the confusion. We have added a more detailed explanation at the beginning of section 3 for clarification.
>     * The context distributions $\pi(\boldsymbol c|o)$ and $\pi(o)$ are required in order to allow the agent to implement its own curriculum for each components (by learning $\pi(\boldsymbol c|o)$). We hence need to define the gating distribution $\pi(o|\boldsymbol c)$ via Bayes rule instead of explicitly modeling the gating distribution $\pi(o|\boldsymbol c)$ as a parametric model, where such a curriculum can not be implemented (at least not as straightforwardly). Note that $\pi(\boldsymbol c)$ is not equal to $p(\boldsymbol c)$ as the agent can choose to ignore context regions if there is no valid component for this region. Hence, for each component, the contexts are sampled from its corresponding normal distributed $\pi(\boldsymbol c|o)$ and thus, it is ensured that the components only see contexts from a local context region, which allows each component to specialize. As a result, this model resembles our desiderata: i) training local experts ii) each component can adjust its curriculum.

---

> > ### Author Response · Authors · 2021-08-30
> > **Reply to Reviewer Kbdi continued**
> >
> > * *"I don't understand why an entropy bonus is useful in ..."*
> >     * Beyond the known benefits of better exploration, in the context of versatile skill learning, the entropy bonus leads to more versatile skills (see also comparison to HiREPS in Section 4.1). I.e., our approach aims to learn all possible theta vectors that lead to a "good reward" solution (depending on the tradeoff parameter alpha). Hence, if some dimensions of the theta vector are irrelevant for the resulting behavior of the agent, the algorithm will increase the variance of this dimension to maximize entropy. Yet, this does not harm performance (as otherwise it would not be irrelevant) and is also avoided by using motion primitives as policy representation as these primitives typically provide a concise description of the motion without irrelevant parameter dimensions. Without the entropy bonus, the algorithm would concentrate on finding a single solution per context (the optimal one) and prune all remaining solutions.
> >
> > * *"I don't see why pi(c|o) or pi(o) (really the probabilities, that are only under indirect control by selecting pi(o|c)) are useful. The difference ..."*
> >     * One key element of our approach is the assumption that $\pi(\boldsymbol c) = \sum_o \pi(\boldsymbol c|o)\pi(o)$ is indeed *under control* of the agent, i.e., the agent can, during training, choose the contexts and learn its own curriculum for choosing $\boldsymbol c$. This is required to allow each component to specialize in a local context region. The goal of  $\pi( \boldsymbol c)$ is not to estimate the real density of the context distribution, i.e., $p(\boldsymbol c)$, as the agent can choose to ignore parts of the context space if there is no valid component covering this space (these ignored regions of the context space will be covered later on by newly added components. This is ensured by the augmented reward function that emerges from the lower bound, see ablation study in Section 4.1). $\pi(\boldsymbol c)$ rather models in which part of the context space the agent is performing well and in which part it isn't. The meaning of $\pi(\boldsymbol c|o) $ is similar -- each component can choose its own curriculum, i.e., which contexts should be used to update the component. Note that during evaluation, we use the given context distribution $p(\boldsymbol c)$ which is specified by the environment.
> >         We are sorry for not making that sufficiently clear in the text and hope that this is now more understandable in the updated version.
> >     * We worked out Section 3.1 in more detail and tried to give an intuitive meaning of the individual terms appearing there.
> > * *"The distinction between trial-based and time-step-based learning algorithms ..."* and *"For parametric component policies, I don't think..."*
> >     * We thank the reviewer for pointing out the unclear statement. We have added a more detailed explanation in the related work part.
> >     * We followed the definition of step-based and episodic approaches as defined in [3], where the episodic policy search methods update their search distribution parameters after gathering experiences at the end of an episode. This different update allows considering Non-Markovian reward functions (as they are used in the experiments of this paper), which is against the definition of step-based approaches.
> >     * In order to show the benefits of both approaches in different settings, we have conducted an experiment on a 5-Link reacher task (a more complex version of the 2Link environment from OpenAI Gym) with a Markovian reward and compared our method to PPO. The results are reported in the Appendix. In general, episodic policy search methods perform exploration in the parameter/weight space $\boldsymbol \theta$ rather than in the raw action space a. These kind of exploration strategies are more successful in sparse-like settings, where the cost is compiled of torque punishments over most of the episode-time and the actual task information (e.g. distance to target reach point) is given in the last step(s) of the episode. This is also a quite common objective in optimal control, where optimal movements are achieved by minimal energy consumption. In the experiments, we show, when our episodic approach is beneficial over PPO and vice-versa.

---

> > > ### Author Response · Authors · 2021-08-30
> > > **Reply to Reviewer Kbdi continued**
> > >
> > > * *"It might be useful to make a connection to "options" as described by Sutton."*
> > >     * We thank the author for pointing this out. We have extended the related work part and included the options framework.
> > > * *"How good are the lower bounds, and how much does it matter?"*
> > >     * The lower bounds are calculated for each component independently during the optimization process. Note that both lower bounds are tight [19], i.e., similar to Expectation Maximization, by improving the lower bounds, we also improve the original objective at each step. Without the lower bound, each component could still be optimized independently but they would not cover distinct context and parameter regions as the objective is missing the augmented rewards. This is demonstrated in the ablation study (section 4.1), where we have analyzed the importance of the augmented rewards.
> > >
> > > * *"The paper promises "unprecedented accuracy and versatility," How is this demonstrated? Too much hype?"*
> > >     * We outperformed SOTA methods (HiREPS, LaDIPS) in terms of quality and versatility of the learned skill library and hence we referred to unprecedented accuracy and versatility. Yet, we agree that more comparisons to step-based approaches would be useful to back this claim and therefore tuned the claim down to "high accuracy and versatility".

---

> ### Comment · Reviewer_Kbdi · 2021-09-03
> **Paper improved, still has major issues.**
>
> I agree the paper has been improved.
> I still don't see a simple intuitive explanation of the approach and why it works in the paper.
> For me to be enthusiastic about this paper, it has to do real tasks on a real robot.
> "I recommend rejecting the paper, but will not argue for my recommendation if the majority of other reviewers have a different opinion."

---

### Author Response · Authors · 2021-08-30
**To all reviewers**

We want to thank all reviewers for their time and valuable feedback. We will try to answer all concerns and questions in the following. Please note, that we had to rearrange and rescale the plots in order not to exceed the page limit due to additional descriptions in the paper.  Also, the equation numbering has changed compared to the previous version. In the following, when referring to equations, figures, or line numbers, we refer to the updated version of the paper. The changed text is marked in blue color. Furthermore, we have conducted a comparison to step-based policy search (PPO) on an additional benchmark, where we also designed different scenarios aiming at demonstrating the advantages of episodic policy search methods and step-based policy search methods. The results are reported in the Appendix C4 (supplementary material). We also run LaDiPS on the Beer Pong and Table Tennis task. The results are in the updated version (Fig. 2a) + Fig. 2b)). A qualitative comparison to learned solutions and different quantities for the Beer Pong task can also be found in the Appendix C.2.1.

---

### Meta-Review · Area_Chair_Wc2b · 2021-08-13

**Recommendation:** Accept (Poster)
**Confidence:** 4

**Metareview:**

Strengths:

- The high level objective of learning skills that encode different ways of accomplishing the same goals was viewed as important.

- The work appears to be technically sound, leveraging max ent policy search and regularization strategies from variational inference. The work extends existing policy search approaches to a multi-modal context in a technically sound way.

- The simulated evaluation of the method on Beer Pong and Ping Pong demonstrated that the method can work in simulation.

Weaknesses:

- Most reviewers had a hard time understanding the development in sect 3. In particular, the development from eq 2 to eq 3 was hard to understand. Some reviewers suggest adding intuitive descriptions of the various terms in eq 3. The use of pi(c) as a variational distribution in eq 2 was unclear (if that’s what it was). Generally, I think there is a lot of “inside baseball” drawn from the policy search community and the variational inference community that makes it hard for many people to parse what is going on here.

- There were questions about the significance of the experimental results relative to the general state of the art. The work would be strengthened by: 1) evaluating the method in simulation on many more tasks; and/or 2) performing experiments on real robots. There was also a question about whether HiREPS was the right baseline for this approach. Additional baselines would help.

- There were questions about sample efficiency resulting from the fact that experiences are not shared between components for the purposes of policy learning.

Post Rebuttal:

The authors have significantly improved the clarity of the described method. This was the primary criticism and multiple reviewers now feel positively after seeing this change.

---

> ### Author Response · Authors · 2021-08-30
> **Reply to the Meta-Reviewer**
>
> We thank the meta-reviewer for the comments. In the following, we want to answer each point individually.
>
> * *"Most reviewers had a hard time understanding the development in sect 3. In particular, ... "*
>
>     * We added a more detailed and intuitive description in Section 3 and reformulated some of the objectives (especially the objective in Equation (4)) to make it easier to understand the steps in between. Furthermore, we made clear that not the marginal context distribution $\pi(\boldsymbol{c}) $ is the variational distribution, but rather the responsibilities introduced in Section 3.2. Please note that the equations 2 and 3 are now numbered as equations 3 and 4 in the newly uploaded version.
>
>
> * *"There were questions about the significance of the experimental results ..."*
>
>     * We agree that evaluating the method on more tasks is valuable. We added a comparison on a standard benchmark task (5Link-reacher) with step-based policy search (PPO) for different reward sparsity scenarios (see supplementary material). Relating to the concerns of reviewers regarding the differences of episodic vs. step-based policy search methods, we also have explicitly mentioned the different benefits of both strategies. We also want to emphasize that we have consciously chosen the environments to be challenging and benefit from versatility as there are explosively multiple solutions available which are not so obvious for many other standard simulation benchmarks. The provided Beer Pong and Table Tennis tasks provide challenging benchmarks, since i) the underlying dynamics are more realistic than commonly used Mujoco environments such as in OpenAI gym, ii) we learn and discover skills from scratch (i.e. no demonstrations) iii) we consider a rather high dimensional context space, iv) we learn multiple solutions.
>     * We agree that real robot experiments would strengthen the paper. However, access to our labs was limited in the past due to the given situation.
>     * We have chosen HiREPS as the baseline, since i) it is an episodic policy search method and thus can deal non-markovian rewards as it is used in our experiments, ii) it is a hierarchical policy search method that allows learning versatile skills, and iii) as our method, it does not make any assumption/requirement on how many components should be at least active. However, We have added LaDiPS to the Beer Pong and Table Tennis task and report the results in Fig. 2 a) and Fig. 2b). For the Beer Pong task we have done a comparison to learned versatile solutions and further report different quantities in terms of versatility in the Appendix C.2.1.
>
> * *"There were questions about sample efficiency resulting ..."*
>     * Currently experience sharing between components is not implemented but can be easily added similar as in LaDIPS. LaDIPS, therefore, outperforms our method in terms of sample efficiency on the Beer Pong task (see plot Figure 2 a)). Yet, we still have a large margin over LaDIPS in terms of versatility (see Appendix C.2.1) and quality of the learned mixture of expert model, as we are able to achieve a higher end-reward. Hence,  we consider experience sharing as an important extension and therefore plan to address this point in future work.

---

### Decision · Program_Chairs · 2021-09-13

**Decision:**

Accept (Poster)

**Comment:**

Strengths:

- The high level objective of learning skills that encode different ways of accomplishing the same goals was viewed as important.

- The work appears to be technically sound, leveraging max ent policy search and regularization strategies from variational inference. The work extends existing policy search approaches to a multi-modal context in a technically sound way.

- The simulated evaluation of the method on Beer Pong and Ping Pong demonstrated that the method can work in simulation.

Weaknesses:

- Most reviewers had a hard time understanding the development in sect 3. In particular, the development from eq 2 to eq 3 was hard to understand. Some reviewers suggest adding intuitive descriptions of the various terms in eq 3. The use of pi(c) as a variational distribution in eq 2 was unclear (if that’s what it was). Generally, I think there is a lot of “inside baseball” drawn from the policy search community and the variational inference community that makes it hard for many people to parse what is going on here.

- There were questions about the significance of the experimental results relative to the general state of the art. The work would be strengthened by: 1) evaluating the method in simulation on many more tasks; and/or 2) performing experiments on real robots. There was also a question about whether HiREPS was the right baseline for this approach. Additional baselines would help.

- There were questions about sample efficiency resulting from the fact that experiences are not shared between components for the purposes of policy learning.

Post Rebuttal:

The authors have significantly improved the clarity of the described method. This was the primary criticism and multiple reviewers now feel positively after seeing this change.